# Changing Geographic Ranges of Human Biting Ticks and Implications for Tick-Borne Zoonoses in North America

## Stephen K. Wikel

Department of Medical Sciences, St. Vincent's Medical Center, Frank H. Netter MD School of Medicine, Quinnipiac University, Hamden, CT 06518, USA; stephen.wikel@quinnipiac.edu; Tel.: +1-203-626-9231

**Abstract:** Ticks and tick-borne pathogens are increasing public health threats due to emergence of novel pathogens, expanding geographic ranges of tick vectors, changing ecology of tick communities, as well as abiotic and biotic influences on tick–host–pathogen interactions. This review examines the major human-biting ixodid tick species and transmitted pathogens of North America. Topics addressed include current and projected tick geographic ranges, potential risks for introduction of tick transmitted microbes into those regions, and drivers for these events. Health care providers, public health authorities, and the general public need to be aware of existing, resurging, and emerging tick and tick-borne disease threats. Knowing which ticks and tick-borne pathogens are present is foundational to understanding and responding to these threats. Dominant tick species and pathogens remain major foci of research, while limited attention has been directed to other human-biting ticks for decades, resulting in questions about current distributions, population sizes, and diversity of infectious agents they are capable of transmitting. Significant threats due to invasive ticks are considered. Recommendations are made for establishment of a sustained North America network for surveillance of ticks, characterization of their microbiomes and viromes, and for support of tick and tick-borne disease ecology research.

**Keywords:** ticks; tick-borne diseases; emerging and resurging pathogens; zoonoses; vector ecology; geographic range; public health importance

## 1. Introduction

Ticks and tick-borne pathogens are persistent and increasingly challenging global public health threats due to expanding geographic ranges, emergence of previously unrecognized tick transmitted infectious agents, and complex dynamic interactions among abiotic and biotic factors that influence the tick–host–pathogen triad [1–4]. Among arthropod vectors, ticks transmit the greatest diversity of infectious agents to humans and livestock [5]. Within the United States, 75 percent of reported vector-borne human infections are attributed to ticks [6,7]. Tick and tick-borne pathogen range expansions are also occurring in Canada [8,9]. Reflecting changing disease patterns and greater diversity of pathogens, forty percent of tick-borne infectious agents in the United States were first recognized as causing human disease during the years since 1980 [10]. The timeline for discovery of tick-transmitted infectious agents in the United States as reported by Eisen and Paddock [11] notes that some microorganisms were identified in ticks prior to their association with human disease, whilst others were linked to human disease prior to knowing they were tick-transmitted. Most notably, *Ixodes scapularis* has received greater attention over the past decades due to increasing importance of multiple transmitted pathogens and the increasing incidence and geographic range expansion of Lyme disease cases. Simultaneously, relatively limited attention has been focused on some other human-biting tick species over the years, resulting in questions about their current distributions, population sizes, ecology, and diversity of infectious agents they transmit. Introduction of exotic tick species and infectious agents they may transmit remains both a human and veterinary public health

concern. Sustained national tick and tick-borne pathogen surveillance programs are needed that provide timely, accessible data that can inform healthcare providers, public health officials, and the general population about potential disease risks in their communities. Ticks and the pathogens they transmit are increasingly transboundary. Therefore, establishing integrated North American surveillance initiatives would be a logical strategy to protect human health and livestock biosecurity.

Human tick-borne diseases are zoonoses [10]. Wildlife species are the origin or reservoir of tick-transmitted infections of humans, companion animals, and livestock [12,13]. In addition to their reservoir roles, wildlife can also serve as hosts that amplify tick populations [4]. Knowledge of the biology of wildlife reservoirs/tick hosts is important for understanding enzootic cycles of tick-borne pathogens, tick population density changes, tick range expansion and contraction, human disease risk, and development of prevention, management, and control measures. One Health integrated approaches can provide valuable insights for assessing disease burdens; establishing surveillance networks to detect infections in humans, reservoir animal species, livestock, wildlife, and arthropod vectors; strengthening laboratory capacity to diagnose infections; and developing multisectoral response capabilities to mitigate, control, and prevent infections [14,15].

Tick-borne viruses, bacteria, and protozoa of currently unknown human pathogenicity can be identified or predicted by application of increasingly powerful sequencing and bioinformatic analyses of tick microbiomes and viromes [16,17] that result in discovery and characterization of potential emerging or future infection threats [17–20]. A tick-associated microbe discovered in this manner that was subsequently established to be a human pathogen is *Borrelia miyamotoi*, an ixodid tick transmitted relapsing fever spirochete [21]. This emerging pathogen initially isolated from *Ixodes persulcatus* is transmitted in North America by *Ixodes scapularis* and *Ixodes pacificus* [6,22]. Incidence of *Borrelia miyamotoi* in human-biting ticks in the United States during the period 2013-2019 was 0.5% to 3.2%, with 59% of those positive ticks also carrying another human pathogen, predominantly *Borrelia burgdorferi* [23]. The power of this approach was also evident in high-throughput sequencing of viromes of field-collected *Amblyomma americanum*, *Dermacentor variabilis*, and *Ixodes scapularis* that resulted in identification of 24 novel viruses [24]. Well-established tick populations can have previously unappreciated vector potential. *Rhipicephalus sanguineus* emerged as a vector of *Rickettsia rickettsii*, a Rocky Mountain spotted fever causative agent, in the southwestern United States [25], and it is an important vector of this pathogen in Mexico [26]. In addition to disease transmission from established tick vectors, ongoing concerns are health and biosecurity threats due to introduction of exotic tick species, such as the recently recognized and expanding presence of *Haemaphysalis longicornis* in the United States [27].

Public health impacts of all tick-transmitted infections remain largely unquantified [28]. However, direct effects of tick infestation and tick-borne infectious diseases have broad economic and societal repercussions [28]. Ticks and tick-borne infections are a significant concern for livestock producers and food security. Globally, 80 percent of cattle are at risk of tick infestations that result in decreased weight gain, reduced milk production, hide damage, and tick-borne infections, resulting in an estimated annual cost of USD 22 to USD 30 billion dollars [29]. Among human tick-borne diseases, Lyme disease is the most commonly reported tick transmitted infection in North America [30]. Thus, it has become a major research focus since its discovery. During the period from 2010 to 2018, yearly estimated frequency of human Lyme borreliosis in the United States was 476,000 cases [31]. Annual economic burden of Lyme borreliosis is estimated to range from USD 786 million [32] to potentially USD 1.3 billion [33].

## 2. Ticks and Tick-Borne Pathogens: Change Is Normal

Cases of tick-borne bacterial and protozoan diseases more than doubled in the United States during the period of 2004 and 2016 [7]. Expansion of tick geographic ranges have been occurring since the mid-twentieth century [2,34]. Bishopp and Trembley in 1945 [35] published an extensive review of the distribution of tick species in North America based upon 35 years of specimens collected and submitted by multiple contributors and an extensive review of the literature. This review provided both foundational material for current distribution maps of multiple tick species and, when compared with current distribution maps for important human-biting ticks in North America, reflects the significant changes, for some species, in geographic distributions over the subsequent years. Although published 77 years ago, Bishopp and Trembley [35] along with the extensive work by Gregson [36] remain as valuable baseline records of tick geographic distributions and host species in the United States and Canada, respectively.

Climate and environmental changes are drivers that influence the incidence of tick-borne diseases [1,37–40]. Ticks are susceptible to the environment in multiple interconnected ways. The tick life cycle involves significant free-living periods of months to years that are interspersed with intervals of host attachment and blood feeding [2,41]. Free-living ticks are susceptible to climate variations that can impact survival, spatiotemporal ranges, and transmitted pathogens in ways unique from other vector arthropods [42,43].

Tick biology and ecology are influenced by variations in temperature, humidity, soil moisture, vegetation, leaf litter, shade, and availability of host species [2,44,45]. Changing climate is resulting in a warmer environment, drier soils, vegetation stress, greater aridity, and river flow declines in North America [46]. Increasing temperatures will continue to be a determinant of changing tick population sizes and geographic distributions [1,45]. Decreased rainfall and resultant drier environment could increase tick mortality [43]. However, the influence of rain events of increased intensity and flooding on tick ecology are basically unknown. Warmer temperatures can increase both the seasonal duration of tick and human activities that result in increased potential for exposure to ticks and pathogen transmission [47]. Increased temperatures also can expand the habitat ranges of tick-transmitted pathogen reservoirs and tick-amplification host species [9]. In response to environmental warming in the northern hemisphere, tick geographic ranges are anticipated to both expand northward and to higher altitudes whilst potentially accompanied by contractions from subtropical and tropical ranges [1,2,45,48].

Factors influencing geographic range and population size are dynamic and dependent upon the complex interplay among micro and macro climate variations, vegetation patterns, land use, land fragmentation, habitat or landscape modification (agricultural, residential, recreational), host animal diversity (domestic, wild and exotic species), human behavior, travel, commerce, economics, government policies, human and animal population growth, population movement, and evolutionary changes in ticks and tick-borne pathogens [1,2,9,12,14,45,49]. Each of these continually evolving factors, to differing degrees, influences tick and pathogen ecology, enzootic cycles of tick-transmitted infectious agents, disease incidence, epidemiology, and selection of appropriate control approaches [2,3,40].

These climate changes impact animal movement and thus geographic distribution and availability of both tick hosts and pathogen reservoirs, resulting in changes of foci where tick-borne diseases occur [2,9,38,40,50]. An additional consideration is the impact of climate variations on the behaviors and routes of migratory birds, an important factor in the distribution of ticks and pathogens into areas where they might not have previously occurred [51]. A comprehensive study at multiple sites in eastern Canada established the role of migratory birds in both introduction and northward range expansion of *Ixodes scapularis* and the associated human pathogens, *Borrelia burgdorferi* and *Anaplasma phagocytophilum*, transmitted by this tick [52]. Landscape determinants impacting bird and mammal movement are important determinants in the northward expansion of *Ixodes scapularis* in Ontario and Quebec [53]. On the Canadian prairie, the range of *Dermacentor variabilis* is expanding northward and westward and now overlaps with *Dermacentor andersoni* in Saskatchewan,

where three bird and multiple mammalian species are infested with different life-cycle stages of these three host tick species [54].

Tick establishment in new regions is accompanied by the introduction of the tick-borne pathogens for which that tick is a competent vector [3,55]. Physicians, veterinarians, public health workers, and the general public need to be made aware in a timely manner of these potential threats as they emerge in a new area. Knowing where ticks and tick-borne pathogens are present is foundational to understanding the health threats they pose. Currently, data and maps describing occurrence and distribution of pathogens in field-collected ticks in the United States are limited, fragmented, and in many instances not published [11]. Surveillance of ticks and tick-borne pathogens provides essential information needed to inform medical care providers, public health officials, and the public of disease risks and to determine measures that may be utilized as appropriate vector and disease-control interventions [11,55–57]. Geographic level of data resolution across the range of a tick species is important for knowing variations of risk and thus tailoring responses based upon disease threat in a specific area [58,59]. Control measures are dependent upon the different populations and balance among tick species present within a region [60]. Surveillance of ticks and tick-borne pathogens has to be an ongoing process. A "one and done" approach to surveillance will only provide a snapshot in time that will very likely become obsolete based upon the changes in tick ranges that have been observed over the past two decades. This process will require both political will and financial commitment necessary to detect, inform those at risk, and respond with appropriate interventions.

### 3. *Ixodes scapularis*

Significant geographic range expansions are occurring among multiple human-biting tick species of North America. *Ixodes scapularis* receives the most attention due to its role, along with *Ixodes pacificus*, as a primary vector of Lyme borreliosis, the most frequently reported tick-borne infection in the United States [30,61]. *Ixodes scapularis* is undergoing changes in ecology concomitant with its emergence as a vector of multiple pathogens and expansion of geographic range [6].

The established geographic range of *Ixodes scapularis* in the United States doubled to 44.7 percent of counties during the period 1996 to 2016 [34] with further expansion in the upper Midwest during 2016 to 2019 [45]. The greatest expansions occurred in the northeastern and north central states [34]. *Ixodes scapularis* is also more northerly established and expanding its range at a rate of approximately 46 km per year in Ontario, Canada [62]. As tick ranges expand, so do the risks and incidences of diseases. *Borrelia burgdorferi* infections are emerging in Canada with a dramatic increase in cases in Ontario and Quebec [62,63]. As anticipated, *Babesia microti* infection is both acquired and increasing in prevalence in Canada [64,65]. Further geographic range expansion of *Ixodes scapularis* is anticipated to continue as it reclaims its historical range over eastern and midwestern North America [6]. This expanding range of *Ixodes scapularis* in the United States and Canada will contribute to an increasing number of Lyme borreliosis and other *Ixodes scapularis* transmitted pathogen infections with greater social and economic impact.

While *Ixodes scapularis* has expanded its range, *Ixodes pacificus* has maintained a steady range along the Pacific coast from California to Washington and in the less-arid regions of Arizona, Nevada, and Utah [34]. *Ixodes pacificus* and *Borrelia burgdorferi* are present in southern British Columbia [66]. Gregson [36] reported multiple records of *Ixodes pacificus* in British Columbia during the early to mid-20th century.

## 4. *Ixodes scapularis*- and *Ixodes pacificus*-Transmitted Pathogens

The majority of human tick-borne infections within the United States are those transmitted by *Ixodes scapularis* and *Ixodes pacificus* [6,30]. *Borrelia burgdorferi* is estimated to account for an annual incidence of 476,000 cases [31]. The annual economic burden of Lyme borreliosis is estimated to range from USD 786 million [32] to potentially USD 1.3 billion [33]. *Ixodes scapularis* was elevated to an important human-biting tick after 1970, when it was found to be a competent vector of *Babesia microti* and over the subsequent fifty years to be a vector of *Borrelia burgdorferi*, *Anaplasma phagocytophilum*, Powassan virus, *Ehrlichia muris eauclarensis*, *Borrelia miyamotoi*, and *Borrelia mayonii* [6,10].

Recently, Fleshman et al. [67] reported distributions of *Ixodes scapularis*- and *Ixodes pacificus*-transmitted human pathogens for the contiguous United States at the spatial resolution of individual counties. Tick vectors occurred over a wider distribution than the human pathogens for which they are competent vectors. *Borrelia burgdorferi* was detected in 30 of 41 states, whilst *Borrelia mayonii*, also a Lyme disease agent, occurred in one county in both Indiana and Michigan [67].

Lyme borreliosis is the topic of comprehensive reviews [61,68] as well as ones that are more focused upon epidemiology [30], diagnosis [69–72], clinical manifestations and treatment [73–75], pathogenesis [76], and the challenges posed by post-treatment syndrome and chronic Lyme disease [77,78]. Significant advances have been made in characterizing immune responses to *Borrelia burgdorferi* infection and the multiple pathways of spirochete evasion of both host innate and adaptive immune defenses [79–84].

*Borrelia burgdorferi* spirochete molecular adaptations are determinants of tick vector colonization that include midgut binding, microbiome modulation upon infection, translocation across the gut to the hemolymph, adapting to tick defenses, salivary gland infection, and adaptations associated with the transition from the tick to the mammalian host tissue environments [85–88]. Changes occur in *Borrelia burgdorferi* numbers in tick tissues when an infected nymph begins to blood feed. Spirochete numbers in midguts of infected *Ixodes scapularis* nymphs increased six-fold by 48 h of host attachment [89]. Spirochete transmission from infected nymphs to a population of 14 hamsters found that infection occurred in one hamster at 24 h of feeding, five hamsters at 48 h, and thirteen hamsters after 72 h or longer [90]. The likelihood that *Borrelia burgdorferi* spirochetes would be transmitted during the first 24 h of tick feeding is low [91]. Compare this transmission profile with that described below for Powassan virus.

*Borrelia miyamotoi* is present in 21 states and the District of Columbia within the range of *Ixodes scapularis* and present in *Ixodes pacificus* in California, Oregon, and Washington [67]. *Borrelia miyamotoi* is an example of a microbe identified in ticks by molecular methods prior to it being established as a human pathogen [21]. This relapsing fever spirochete, transmitted by ixodid rather than argasid ticks, may cause a range of clinical manifestations that include meningoencephalitis observed in immunocompromised patients [22,92]. *Borrelia miyamotoi* and *Borrelia burgdorferi* occur concurrently across the northern hemisphere, where they share both tick and vertebrate reservoir hosts [93].

*Anaplasma phagocytophilum* occurs in association with *Ixodes scapularis* in 23 northeastern and midwestern states and the District of Columbia and with *Ixodes pacificus* in northern California and Washington [67]. A finding that has implications for the future distribution of *Anaplasma phagocytophilum* is the detection of this infectious agent in field-collected, questing *Haemaphysalis longicornis* in the United States [94]. A meta-analysis of the global human seroprevalence of *Anaplasma phagocytophilum* revealed a pooled seropositivity rate of 8.4 percent with wide variations observed in different studies and regions [95]. An important conclusion derived after analysis of data from 56 studies is that infection surveillance misses mild and asymptomatic infections [95]. Clinical features and pathogenesis of human granulocytic anaplasmosis caused by *Anaplasma phagocytophulum* were reviewed by Ismail and McBride [96] along with emerging *Anaplasma* agents.

*Babesia microti* infection is present in approximately 20 percent of *Ixodes scapularis* nymphs in endemic foci [97,98]. *Babesia microti* infections are endemic in 17 northeastern

and upper midwestern states but not reported from western states [67]. Human babesiosis caused by *Babesia microti* is an emerging infectious disease that is a particular threat to immunocompromised individuals and when acquired by transfusion [99]. Clinical features, diagnosis, and treatment of human babesiosis are reviewed by Vannier et al. [98] and Krause [99]. Significant effort is directed toward development of improved, novel approaches for the diagnosis of early infections, particularly before seroconversion and to distinguish active from prior infections [100].

*Ehrlichia muris eauclairenis* was found in *Ixodes scapularis* from limited foci in both Michigan and Minnesota [67]. This emerging *Ehrlichia* species was first detected during 2009 in a series of four patients in the two states just cited [101]. *Ehrlichia muris eauclairenis* can be transmitted in a cycle between *Ixodes scapularis* and rodents with wide tissue distribution and high virulence for mice and hamsters [102]. *Amblyomma americanum* is not a competent vector for this pathogen [102]. *Ehrlichia muris eauclairenis* was described as a novel subspecies infecting *Ixodes scapularis* and *Peromyscus leucopus* in Minnesota and Wisconsin [103].

Powassan virus infections are infrequent; however, they are recognized as an increasing public health threat [104]. Powassan-virus-infected *Ixodes scapularis* exist as foci in New England, the mid-Atlantic, and upper midwestern states [67]. Incidence of Powassan virus infection was reported to have increased 671 percent during an 18-year period prior to the publication by Fatmi et al. [105] that described Powassan virus as a newly remerging infection. In addition to *Ixodes scapularis*, *Ixodes cookei* is an established Powassan virus vector in the midwestern United States and Canada, and *Haemaphysalis longicornis* is a competent vector in Russia [105]. Consideration should be given to potential for *Haemaphysalis longicornis* to become a vector of this virus in North America, with implications for an expanded geographic range for infections. Interest in Powassan virus is reflected in recently increased surveillance, basic research, and animal model development [106]. Clinical presentation, clinical course, diagnosis, and pathogenesis of Powassan virus infection were recently reviewed [107].

In contrast to other *Ixodes scapularis*-vectored pathogens [91], Powassan virus is transmitted to mice as soon as 15 min of feeding, with maximum efficiency of virus transmission at 180 min [108]. This rapid transmission of Powassan virus post host attachment was confirmed for human clinical cases [109].

Significantly, *Peromyscus leucopus* is a reservoir host for these human pathogens for which *Ixodes scapularis* is a competent vector [10,103,110]. *Ixodes scapularis* nymphs are the source of the vast majority of human infections with *Anaplasma phagocytophilum*, *Babesia microti*, and *Borrelia burgdorferi* based upon seasonal incidence of infections [10]. Co-infection of *Ixodes scapularis* with multiple human pathogens is an established phenomenon [111,112]. A recent prevalence study of *Borrelia miyamotoi* infection of *Ixodes scapularis* revealed that 59 percent of ticks had anywhere from dual to quadruple coinfections [23]. The potential for co-infections should be taken into account in any diagnostic workup in a suspected case of tick bite.

## 5. *Ixodes cookei*

*Ixodes cookei*, a tick occasionally reported to bite humans, occurs in a broad area of eastern North America where it infests a variety of small and midsized mammals and is an established vector of Powassan virus [105,113]. Powassan virus was isolated from adult and nymphal *Ixodes cookei* recovered from skunks and long-tailed weasels in New England [114]. *Ehrlichia muris*-like DNA was detected by molecular methods in *Ixodes cookei* in the northeastern United States, suggesting an enzootic cycle different from that of *Ehrlichia muris eauclairenis* in Michigan and Minnesota and a potential human health threat [115]. Due to the overlapping geographic range of *Ixodes cookei* and *Borrelia burgdorferi*, it is important to note that *Ixodes cookei* is a poor vector of the spirochete causing Lyme borreliosis [116].

*Ixodes cookei* geographic range in the United States extends northward from Tennessee and North Carolina into southern Canada [113]. Modeling of suitable *Ixodes cookei* habitat in Canada includes Nova Scotia, eastern Labrador and Newfoundland, New Brunswick, Quebec, and Ontario [113]. Gregson [36] described *Ixodes cookei* as a common eastern tick in Canada with reported locality records from the following provinces: Prince Edward Island, Nova Scotia, New Brunswick, Quebec, and Ontario. A recent analysis of ticks submitted in a passive tick surveillance program from 2007 to 2015 in Quebec, Canada, revealed that 91.2 percent of submitted ticks that were species other than *Ixodes scapularis* were *Ixodes cookei* [8].

## 6. *Amblyomma americanum*

*Amblyomma americanum* is thought to be more widely distributed in North America than currently realized [117]. *Amblyomma americanum* is an aggressive tick that is an important pest of humans and livestock with larvae, nymphs, and adults readily feeding on large mammals [118]. *Amblyomma americanum* is a competent vector of an increasing number of zoonotic pathogens of medical importance [119]. Contributing to these concerns, *Amblyomma americanum* is expanding northward from its traditionally recognized southeastern United States range into the mid-Atlantic states, New England, and the provinces of Ontario and Quebec in southern Canada [2,63,119–122]. Westward expansion of *Amblyomma americanum* includes the midwestern states of Michigan, Nebraska, and South Dakota, whilst climate-change-induced range contraction could occur along the Gulf coast and lower Mississippi river region [2,120]. Modeling future distribution of *Amblyomma americanum* predicts expansion along the eastern seaboard into the maritime provinces of Canada and further northward in the Upper Midwest with the potential for loss of range in the current southern limits of the species [117]. Significantly, *Amblyomma americanum* is increasing in incidence, while *Dermacentor variabilis* populations decline in regions where both species occur [119,123]. Changes in population balance have significant implications for tick-control measures since specific management strategies differ depending upon the tick species to be controlled [57].

## 7. *Amblyomma americanum* Transmitted Pathogens

*Amblyomma americanum* is a tick whose public health importance has steadily increased due to its role as a vector of both established and newly recognized zoonotic pathogens [118,119]. In addition to *Dermacentor andersoni* and *Dermacentor variabilis*, *Amblyomma americanum* is a vector of *Francisella tularensis*, a causative agent of tularemia [10,124], and the primary vector for *Ehrlichia chaffeensis* and *Ehrlichia ewingii*, largely under-recognized and under-reported ehrlichioses that are now present in regions into which this tick recently expanded its range [10,118,119]. White-tailed deer are reservoir hosts for both of these *Ehrlichia* species, and it is important to note that larvae, nymphs, and adults of *Amblyomma americanum* readily acquired blood meals from white-tailed deer [118,125,126]. *Amblyomma americanum* is capable of acquiring, maintaining, and transmitting *Rickettsia rickettsii*, a causative agent of Rocky Mountain spotted fever, under laboratory conditions [127].

Expanding number of microbes transmitted by *Amblyomma americanum* increases the public health importance of this tick. Panola Mountain *Ehrlichia*, an organism of unclear human pathogenicity, is transmitted by *Amblyomma americanum*, and it infects deer [128,129]. Heartland virus transmitted by *Amblyomma americanum* is an emerging human infection that is closely related to severe fever with thrombocytopenia syndrome virus, an emerging tick-borne hemorrhagic fever with a high fatality rate that is endemic in central and eastern China, Korea, and Japan [130–133]. White-tailed deer are implicated as a Heartland virus reservoir due to their widespread antibody seropositivity for this virus [131,134]. Bourbon virus is another emerging pathogen transmitted by *Amblyomma americanum* [135–137]. Human hypersensitivity to saliva proteins of *Amblyomma americanum* introduced during blood feeding can stimulate development of alpha-Gal (galactose-$\alpha$-1,3-galactose) syndrome, a red meat allergy that is a unique food allergy of increasing frequency [138–140].

### 8. *Amblyomma maculatum*

Geographic range of the Gulf Coast tick, *Amblyomma maculatum*, is expanding in multiple directions [141]. Historical range of this species in the United States was the southeastern states bordering the Gulf of Mexico to the Atlantic coast of South Carolina with a 150 miles inland extension along that range [142,143]. *Amblyomma maculatum* moved northward into North Carolina, Virginia, eastern Maryland, and Delaware, accompanied by expansion further inland from the coast and into Kentucky, Tennessee, Arkansas, Oklahoma, and Kansas [2,11,143]. Recently, populations of this tick were found in Illinois [144], Arizona, and New Mexico [145]. The northward expansion of *Amblyomma maculatum* continues with recent detection of an established population in Connecticut [146], New Jersey, and New York City [147].

### 9. *Amblyomma maculatum* **Transmitted Pathogens**

*Amblyomma maculatum* is a competent vector for *Rickettsia parkeri*, a spotted fever group rickettsia that was recognized as a human pathogen in 2002 [148]. This rickettsia was described decades prior to it being established as a cause of rickettsiosis [149,150]. *Rickettsia parkeri* has been detected in *Amblyomma maculatum* collected over much of the geographic range of this tick, including the northern-most reported established population in Connecticut [10,146]. *Rickettsia parkeri* are present in an established population of *Amblyomma maculatum* in New York City [147,151]. Although a vertebrate reservoir(s) is not determined, *Rickettsia parkeri*-infected *Amblyomma maculatum* have been collected from white-tailed deer, feral swine, and other wildlife species [10]. These same wildlife species along with migratory birds and movement of infested cattle are linked to *Amblyomma maculatum* range expansion [142].

### 10. *Dermacentor andersoni* **and** *Dermacentor variabilis*

Open shrubby, semiarid grasslands are habitat for *Dermacentor andersoni*, the Rocky Mountain wood tick, whose range is intermountain and Rocky Mountain western North America from southern British Columbia, Alberta, and Saskatchewan in Canada to northern New Mexico, Arizona, eastern California, Oregon, and Washington in the United States [10,36,152,153]. Significantly, the geographic range of *Dermacentor andersoni* has remained essentially stable since 1932 [152]. Big sagebrush is associated with a greater risk of exposure to *Dermacentor andersoni* adults, and grass was a favored substrate for host seeking ticks [153].

*Dermacentor variabilis* occurs in 45 states, particularly in old fields and woodlands, east of the Rocky Mountain region and with established populations in California, Oregon, Washington, and Idaho [10,154,155]. The geographic range of *Dermacentor variabilis* is expanding westward in the United States. The western distribution of *Dermacentor variabilis* into the Rocky Mountain region of the United States might be changing based upon the observation that it is the predominant tick removed from cats and dogs rather than *Dermacentor andersoni* [156]. While this might reflect simply host preference, the distribution of *Dermacentor variabilis* merits further investigation. Recent molecular analyses of ticks from the eastern and western sides of the described range support the concept that *Dermacentor variabilis* consists of two species with the western populations comprised of a new species, *Dermacentor similis* n.sp. [157].

A survey of the geographic range of *Dermacentor andersoni* in Alberta, Canada, revealed decreasing numbers of ticks with northerly sampling to 51.6° N with greatest abundance in dry mixed grass and montane regions [158]. Distribution of *Dermacentor andersoni* has remained essentially stable in Saskatchewan, Canada, when compared with 1960s data [54]. In contrast, *Dermacentor variabilis* range expanded westward and northward in Saskatchewan, resulting in a zone of overlap approximately 200 km wide with *Dermacentor andersoni* that was not evident in earlier data [54].

## 11. *Dermacentor andersoni-* and *Dermacentor variabilis-*Transmitted Pathogens

Causative agent of Rocky Mountain spotted fever, *Rickettsia rickettsii*, is linked to transmission by both *Dermacentor andersoni* and *Dermacentor variabilis* [10]. The vector roles of these two ticks require further study. *Rickettsia rickettsii* infection is associated with tick mortality raising the question as to the actual prevalence of infection of this tick species in nature and pathogen transmission rates [159]. Similar questions arise regarding the Rocky Mountain spotted fever vector role of *Dermacentor variabilis* due to the high frequency of non-pathogenic spotted fever group rickettsia found in ticks removed from infested humans [160]. Rocky Mountain spotted fever was the subject of comprehensive reviews [161–163].

Both *Dermacentor andersoni* and *Dermacentor variabilis* are vectors of *Francisella tularensis*, causative agent of tularemia [10,124]. Common North American tick vectors of *Francisella tularensis* are *Amblyomma americanum*, *Dermacentor andersoni*, *Dermacentor occidentalis*, and *Dermacentor variabilis* [124]. Ecology of *Francisella tularensis* was recently reviewed [164]. *Dermacentor andersoni* is the primary vector of Colorado tick fever virus to humans in the range of this vector in Canada and the United States [10,165]. Colorado tick fever has been the topic of several reviews [165–167].

Tick paralysis is characterized by an acute ascending flaccid paralysis of humans and other vertebrate species that is a relatively uncommon occurrence induced by feeding of multiple ixodid and argasid species [168–172]. Commonly encountered North American human-biting tick species that can induce paralysis include *Amblyomma americanum*, *Amblyomma maculatum*, *Dermacentor andersoni*, *Dermacentor variabilis*, *Ixodes scapularis*, and *Rhipicephalus sanguineus* [173]. *Dermacentor andersoni* is the predominant tick associated with livestock and human tick paralysis in the northwestern United States and British Columbia, Canada [174–176]. Significantly, cattle can develop immunity to *Dermacentor andersoni* tick paralysis [177].

## 12. *Rhipicephalus sanguineus*

*Rhipicephalus sanguineus* is the most widely distributed tick species globally [178]. This tick predominantly infests dogs; however, it will also infest a range of domestic and wild hosts, including birds, cats, humans, and rodents [178,179]. In North America, the northern and western extent of *Rhipicephalus sanguineus* populations is reflected in their presence in Alaska [180]. An examination of early Canadian records revealed *Rhipicephalus sanguineus* populations associated with dogs in the provinces of Ontario, Quebec, and Nova Scotia [36]. Analysis of 9423 ticks, other than *Ixodes scapularis*, examined as submitted specimens in passive survey in Quebec revealed 4.0 percent were *Rhipicephalus sanguineus*, and 4.1 percent were *Dermacentor variabilis*, the American dog tick [8]. Factors contributing to successful adaptation of *Rhipicephalus sanguineus*, a hunter tick, to widely diverse habitats include the ability of all life cycle stages to obtain blood meals from the same host species and off host life stages that are adapted to living indoors, such as kennels, as well as outdoors in environments with sheltering features, such as rock walls; and, although a three-host tick, *Rhipicephalus sanguineus* is capable of completing four generations a year, dependent upon host availability and environmental conditions [178].

*Rhipicephalus sanguineus* was the unexpected vector of human *Rickettsia rickettsii*, or Rocky Mountain spotted fever, infections in eastern Arizona [181]. Both *Rickettsia rickettsii* and *Bartonella henselae* were detected in questing adult *Rhipicephalus sanguineus* collected in southern California [182]. Globally, *Rhipicephalus sanguineus* is a competent vector of multiple additional pathogens of medical and veterinary importance that include: Mediterranean spotted fever (*Rickettsia conorii*), Q fever (*Coxiella burnetii*), equine piroplasmosis (*Theileria equi*), equine babesiosis (*Babesia caballii*), bovine anaplasmosis (*Anaplasma marginale*), canine babesiosis (*Babesia canis*, *Babesia gibsoni*), canine hepatozoonosis (*Hepatozoon canis*), and canine monocytic ehrlichiosis (*Ehrlichia canis*) [183]. Warmer temperatures result in more rapid attachment and feeding by *Rhipicephalus sanguineus* [178].

### 13. *Haemaphysalis longicornis*: Invasive Vector Tick

*Haemaphysalis longicornis*, the Asian longhorned tick, is native to East Asia [184] and extended its range into Australia, New Zealand, and several Pacific Islands [185,186]. Initial detection of a free-living population of *Haemaphysalis longicornis* in the United States occurred on sheep in New Jersey in 2017 [187]. *Haemaphysalis longicornis* has spread to at least 12 predominantly eastern states within the United States [188]. Examination of archived tick specimens revealed that *Haemaphysalis longicornis* was present in the United States in 2010 if not before [189]. Identification of three mitochondrial DNA *cox* 1 haplotypes among field-collected *Haemaphysalis longicornis* populations across three states indicates that at least three unrelated female ticks were introduced into the United States [188]. Phylogeographic analysis indicates that the introduced ticks were from northeast Asia [188].

Reproductive characteristics contribute to the potential for geographic spread for this tick species. One form of reproduction is bisexual populations that are essentially a 50:50 ratio of females:males, and the second form of reproduction is parthenogenetic, with the vast majority of adults possessing a female morphology [184,188]. Parthenogenetic reproduction results in large numbers of progeny from a single female [189]. Medium- and large-sized vertebrates but not small mammals or birds have large numbers of infesting *Haemaphysalis longicornis* in nature [190]. This parthenogenetic pattern is the reproductive form present in the United States [188].

Models developed to project the geographic range of *Haemaphysalis longicornis* in North America indicate suitable habitat across the eastern and the midwestern United States from the Gulf of Mexico coast into southern Canada as well as a narrow region along the Pacific coast from southern California into British Columbia [117,191]. The potential exists for *Haemaphysalis longicornis* to establish in the maritime provinces of Canada [117] as well as in southern Quebec, Ontario, and Manitoba [192]. Ability of this tick to infest a wide variety of vertebrate hosts means that the potential for movement and spread of range are significant [117]. *Haemaphysalis longicornis* has the ability to adapt to a wide range of climate conditions and habitats such as woodlands, open pastures, and shrubby brush [186]. *Haemaphysalis longicornis* will expand over suitable range and continue to spread northward and westward, as is generally predicted for tick species in a warming northern hemisphere [122].

*Haemaphysalis longicornis* is a competent vector of an array of pathogens of human and veterinary public health importance [117]. Severe fever with thrombocytopenia syndrome is a potentially lethal bunyavirus for which *Haemaphysalis longicornis* is both a vector and reservoir in Asia [130,133]. Although severe fever with thrombocytopenia syndrome virus has not been detected in ticks in North America, the closely related Heartland virus occurs across the central Midwest into southeastern states where *Haemaphysalis longicornis* occurs [117,191,193]. Potential exists for significant emergence of Heartland virus as a public health threat as the enzootic cycle becomes more widely established, and this tick bites more humans. *Haemaphysalis longicornis* transmits Heartland virus transovarially, and experimentally infected ticks transmit the virus to mice [194]. Bourbon virus, an emerging Thogotovirus, was detected in field-collected *Haemaphysalis longicornis* in Virginia, indicating a possible additional tick vector for this pathogen [195]. *Haemaphysalis longicornis* is vector of the virulent veterinary pathogen *Theileria orientalis* genotype Ikeda strain that it is being transmitted to livestock in Virginia [196]. This finding represents an emerging threat to cattle producers in North America.

### 14. Tick and Pathogen Control

The examples just described are characteristic of the broad phenomena of tick species range expansions, population growth, and emergence of tick transmitted pathogens that have significant implications for human and veterinary health. Nowhere is the impact of these changes more challenging than in the efforts to manage, control, and suppress ticks and tick-borne diseases. Effective control of tick vectors and tick-borne pathogens of humans remain vexing and evolving challenges [56,57].

Although the focus of this review is human health, the problems encountered over decades by cattle producers are illustrative of the difficulties in achieving effective tick control [197–199]. Acaricides have remained a mainstay of tick control on livestock since the introduction of arsenic tetraoxide just prior to the start of the twentieth century [199]. With the first report of tick resistance to arsenic as an acaricide in 1936, the ongoing search for new acaricidal compounds and the sequential development of resistance by multiple tick species to each successive generation of acaricides began [198–200]. Acaricide classes synthesized and to which ticks developed recurring resistance include organochlorines, organophosphates, carbamates, formamidines, pyrethroids, macrocytic lactones, neonicotinoids, growth regulators, and phenylpyrazoles [198–200]. Considerable effort has been directed toward defining tick acaricide resistance mechanisms and exploring resistance prevention mechanisms that include addition of enzyme inhibitors as well as use of combinations and rotations of acaricides [199–201]. Research continues to focus on development of novel acaricides that are more selective, less likely to induce tick resistance, and have greatly reduced environmental impact, including the problem of meat and milk residues [202]. Pasture management and reduced acaricide treatments are being incorporated into development of integrated tick management programs for cattle tick control [203–205]. There clearly exists a need for innovative methodologies for the control of cattle ticks that can become an additional tool for use in integrated tick management programs. Anti-tick vaccines are one approach that have proven efficacy for reducing infestations of cattle, and their development remains a significant research focus [206].

Tick bite and tick-borne disease prevention relies largely upon individual decisions to utilize personal protective measures that are the primary manner by which humans protect themselves from these threats [191,207,208]. Those personal protective measures include repellents, protective clothing, tick checks to remove any infesting ticks, avoidance of tick habitat, preventive behaviors, education, and permethrin-treated clothing [207]. Additional risk reduction measures include domestic or wider area environmental modification of tick habitat by landscape modification and application of approved acaricides [207]. Perceived risk is an important factor in individual implementation of personal protection [209].

The problem of significantly increasing numbers and diversity of tick-borne infections in the United States has increased calls for a national strategy to address tick-transmitted disease threats [56]. Concomitant with this initiative is an effort to raise awareness of the need for focused and area-wide integrated tick management programs; increasing incentives to academic and industry to develop, test, and register new tick-control technologies; update strategies to address the increasingly more complex tick and disease threats occurring in a changing landscape; and to expand educational initiatives for both professions and the public [57,210,211]. An issue that will require ongoing research and monitoring is selection of appropriate tick control or suppression methods since approaches differ based upon the different human-biting tick species in an area [57]. Essential to all of this is knowing what the threats are and where they exist.

## 15. Current Situation

Tick geographic range changes require surveillance that is ongoing to capture the dynamic processes that vary for different species. Table 1 provides links to websites that provide information regarding tick species geographic ranges in Canada and the contiguous United States. Table 2 lists major human-biting ixodid ticks of Canada and the United States and the human infectious agents for which their vector competence is established.

**Table 1.** Links to websites in Canada and United States that provide maps of tick distributions based upon recent surveillance data.

| Organization | Website |
|:---:|:---:|
| **Canadian Veterinary Medical Association Tick Talk** Provides information regarding tick species occurring in each province, tick habitat, tick geography, and tick season. | https://ticktalkcanada.com/geographic-expansion/ (accessed on 8 August 2022) |
| **eTick Canada** Map points reflect the different locations where ticks were reported and not the total number of submissions. Only the most recent 20,000 data points are displayed on the map. | https://www.etick.ca/etickapp/en/ticks/public/map (accessed on 8 August 2022) |
| **Government of Canada Lyme Disease Risk Area Maps** | https://www.canada.ca/en/public-health/services/diseases/lyme-disease/surveillance-lyme-disease.html#a4 (accessed on 8 August 2022) |
| *Ixodes scapularis* **Risk Areas in Manitoba, Canada** | https://www.gov.mb.ca/health/publichealth/cdc/tickborne/surveillance.html (accessed on 8 August 2022) |
| **United States Centers for Disease Control and Prevention: Regions Where Ticks Live** Maps show the general distribution of human-biting ticks in the contiguous United States. | https://www.cdc.gov/ticks/geographic_distribution.html (accessed on 8 August 2022) |

**Table 2.** Major human-biting ixodid tick species of Canada and the United States and infectious agents for which their vector capacity is established.

| Human-Biting Tick | Human Infectious Agents Transmitted |
|:---:|:---:|
| *Amblyomma americanum* | *Ehrlichia chaffeensis* *Ehrlichia ewingii* Panola Mountain *Ehrlichia* *Francisella tularensis* Heartland virus Bourbon virus |
| *Amblyomma maculatum* | *Rickettsia parkeri* |
| *Dermacentor andersoni* | *Rickettsia rickettsii* *Francisella tularensis* Colorado tick fever virus |
| *Dermacentor variabilis* | *Rickettsia rickettsii* *Francisella tularensis* |
| *Haemaphysalis longicornis* | Laboratory competence for Heartland virus and Bourbon virus |
| *Ixodes cookei* | Powassan virus |
| *Ixodes pacificus* | *Anaplasma phagocytophilum* *Borrelia burgdorferi* *Borrelia miyamotoi* |
| *Ixodes scapularis* | *Anaplasma phagocytophilum* *Babesia microti* *Borrelia burgdorferi* *Borrelia miyamotoi* *Borrelia mayonii* *Ehrlichia muris euclairensis* Powassan virus |
| *Rhipicephalus sanguineus* | *Rickettsia rickettsii* Q fever |

**Funding:** This research received no external funding.

**Institutional Review Board Statement:** Not applicable.

**Data Availability Statement:** Data sharing not applicable to this review manuscript.

**Acknowledgments:** Appreciation is expressed to Gina Addona, Supervisor, Access Services and Document Delivery, Edward and Barbara Netter Library, Center for Medicine, Nursing and Health Sciences, Quinnipiac University, for her timely and unfailing assistance in obtaining numerous manuscripts needed for the preparation of this review.

**Conflicts of Interest:** The author is Senior Scientist at US Biologic, Inc. Memphis, TN 38103, USA, and he is a stockholder in US Biologic, Inc.

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
