# Peer review of "Changing Geographic Ranges of Human Biting Ticks and Implications for Tick-Borne Zoonoses in North America"

_zoonoticdis, doi:10.3390/zoonoticdis2030013_

Round 1

Reviewer 1 Report

Throughout the entire manuscript, the scientific names are not italicized.  The scientific names in the references are not italicized either.

In the 1st paragraph, can the author find a more recent reference (than #5) regarding the number of agents transmitted by ticks to humans and livestock?

Rickettsia rickettsii is spelled incorrectly towards the bottom of paragraph 4 in the introduction.

The author continues to use complete scientific names (genus and species) when after the first mention of, the first letter of the genus can be used.

Author Response

  1. Scientific names are not italicized throughout text and cited references. Response: All scientific names are now italicized. 
  2. Reference number 5, although published in 2004, remains the most comprehensive citation for tick-borne diseases of human and other vertebrate animal species. Throughout the text citations are provided to the most recent literature addressing relevant tick-borne diseases discussed. As a review focusing on zoonoses, my professional opinion remains that reference number 5 shows the scope of tick-borne disease threats and thus a underpinning for eventual development of a One Health approach to tick-borne diseases that are all zoonoses.
  3.  Spelling error of Rickettsia rickettsii  is corrected. 
  4.  Complete scientific names were used to avoid confusion for readers because of the diverse array of tick species and tick-borne pathogens. I believe this to be particularly important for non-specialist and student readers. They do not need to keep going back in the text to see what the abbreviation means. Response: I am requesting that full scientific names be used. If you and the editors feel strongly that the first letter of the genus be used after the first citation in the text, Return the proof to me and I shall revise accordingly.

Reviewer 2 Report

Manuscript untitled „Changing Geographic Ranges of Human Biting Ticks and Implications for Tick-borne Zoonoses in North America ” presents a large review of actual knowledge about most common ticks redistribution and pathogenic potential.

Article is divided into clear parts: Introduction, Ticks and Tick-borne Pathogens: Change is Normal Ixodes scapularis, Ixodes scapularis and Ixodes pacificus Transmitted Pathogens, Ixodes cookei, Amblyomma americanum, Amblyomma americanum Transmitted Pathogens, Amblyomma maculatum, Amblyomma maculatum Transmitted Pathogens, Dermacentor andersoni and Dermacentor variabilis, Dermacentor andersoni and Dermacentor variabilis Transmitted Pathogens, Rhipicephalus sanguineus, Haemaphysalis longicornis: Invasive Vector Tick, Tick and Pathogen Control, Current Situation.

Tick-borne pathogens as a public health because of novel pathogens, expanding geographic ranges of tick vectors, changing ecology of tick communities, as well as abiotic and biotic influences on tick-host-pathogen interactions.

In next several parts are presented important to human environment ticks and transmitted by them pathogens.

To sum up, I give a positive opinion about manuscript untitled „Changing Geographic Ranges of Human Biting Ticks and Implications for Tick-borne Zoonoses in North America ”.

Author Response

There were no requested revisions. Thank you for your review.